# Modeling and Use of Inter-Criteria Decision Analysis for Selecting Growth Rate Models for Batch Cultivation of Yeast *Kluyveromyces marxianus var. lactis* MC 5

Mitko Petrov 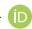

Institute of Biophysics and Biomedical Engineering, Bulgarian Academy of Sciences, 105 Acad. George Bonchev Str., 1113 Sofia, Bulgaria; mpetrov1959@gmail.com

**Abstract:** Ten unstructured models of Monod, Mink, Tessier, Moser, Aiba, Andrews, Haldane, Luong, Edward, and Han-Levenspiel are considered in this paper to explain the kinetics of cell growth for batch cultivation of the yeast *Kluyveromyces marxianus var. lactis* MC 5. For the first time, two independent kinetic models are used to model the process for the two basic substrates—lactose and oxygen. The selection of the most appropriate growth rate models has been made through a new multi-criteria decision-making approach called the Inter-Criteria Decision Analysis (ICDA) method. The application of ICDA to the growth rate of lactose and oxygen alone has shown that there have been many correlations between the studied models. Thus, the models for the growth rate, depending only on lactose, are reduced to one—Monod model and there are two models—Monod and Mink—depending on oxygen only. Separate kinetic process models have been developed for the combination of Monod–Monod and Monod–Mink models. For the first time, in addition to the multiplicative form, the additive form of a specific growth rate has been studied. The comparison of the obtained results has shown that the additive form has shown better results than the multiplicative one. For this reason, the additive form of the Monod–Monod model will be used to model the process.

**Keywords:** modeling; yeast *Kluyveromyces marxianus var. lactis* MC 5; growth rate models; inter-criteria decision analysis; intuitionistic fuzzy sets; index matrix; intuitionistic fuzzy pairs

## 1. Introduction

Cheese is a unique product with a high caloricity and physiological sufficiency. It is conditioned from the high proteins and greases, from the presence of well-assimilated human organism peptides and free amino acids, vitamins, and microelements. The white brine cheese is a rich source of calcium, vitamins A, D, E, and vitamins from the B group, lipids, sodium, and lactose [1].

Strains belonging to the yeast species *Kluyveromyces marxianus* have been isolated from a great variety of habitats, which results in a high metabolic diversity and a substantial degree of intraspecific polymorphism. As a consequence, several different biotechnological applications have been investigated with this yeast: production of enzymes, of single-cell protein, of aroma compounds, and of ethanol (including high-temperature and simultaneous saccharification-fermentation processes); reduction of lactose content in food products; production of bioingredients from cheese-whey; bioremediation as an anticholesterolemic agent and as a host for heterologous protein production [2–6].

There are many methods [7] for multi-criteria decision-making (MCDM). The most commonly used are analytical hierarchy processes [8], elimination and choice expressing reality (ELECTRE) [9], multi-attribute utility theory (MAUT) [10], and the preference ranking organization method (PROMETHEE) [11]. Recently, methods such as intuitionistic fuzzy MCDM [12] and Type-2 neutrosophic number-based multi-attributive [13] have also gained popularity.

In this work, we use a new Inter-criteria Decision Analysis (ICDA) method based on the belief functions to identify and assess the possible degree of agreement (i.e., excess) between some criteria derived from the data (point values). This allows us to remove any redundant criteria or objects of the original MCDM problem and thus solve a simplified (almost) and equivalent MCDM problem faster and at lower computational costs. The ICDA was originally developed by Atanassov et al. [14] based on intuitionistic fuzzy sets [15–18] and indexed matrices [19–21]. The approach uses the concept of index matrix (IM), using, in particular, some of the operations introduced on them, and the concept of intuitionistic blurring, thus giving us the tools to construct an IM of intuitionistic fuzzy pairs [22].

The method is applied for the analysis and modeling of biotechnological processes [23–27] and for evaluating the index of pollution of Struma and Mesta rivers [28,29], etc.

There is no general mathematical model of microbial synthesis due to the extreme complexity and great variety of vital activity of microorganisms, although there are different models of the biotechnological process. Recently, efforts have been made to develop generalized models in the field of biotechnology [30].

An unstructured kinetic model was developed [31] for the serial production of bioethanol from the yeast *Kluyveromyces marxianus* DSM 5422, as well as from renewable sources and agricultural and food products. The experimental data have been collected from multiple fermentations in bioreactors with three different initial concentrations for each substrate. This data has been used to evaluate the parameters and to validate a Monod model of the Haldane type in combination with the term Jerusalimsky for non-competitive inhibition of the product.

In our publications, attention has been focused on modeling the bath process of cultivation of yeast *Kluyveromyces marxianus var. lactis* MC 5 taking into account the mass exchange processes [32–34] in [32] a mathematical model of batch cultivation of yeast *Kluyveromyces marxianus var. lactis* MC 5. In the model of the process, the mass-transfer in the bioreactor for oxygen concentration in the gas phase (GP) and the liquid phase (LP) is based on the dispersion model of the GP. In addition, perfect mixing in LP is included. Nine growth rate models for lactose and oxygen have been studied: Monod, Mink, Tessier, Aiba, Andrews, Haldane, Luong, Edward, and Han-Levenspiel. The models are compared using criteria for identification of kinetics parameters in the models, relative error, and statistic λ. Based on the study (the comparison was made according to the above criteria), it was found that the models of Mink and Haldane, Mink and Luong, and Mink and Han-Levenspiel fit best. The main disadvantage of this study is that the selected combinations of models are made by simply comparing the accepted criteria. Some of the MCDM methods have not been used.

To address this shortcoming in the selection of models in [33], the ICDA was used to evaluate the same models. The application of ICDA has shown that the most suitable for lactose are the models of Monod and Mink. As far as it concerns the growth rate from oxygen, the most suitable ones are the models of Mink, Tessier, and Haldane. In this way, the application of the ICDA has permitted us to investigate the combination of only groups of models—Monod for lactose and Mink, Tessier, and Haldane for oxygen, and Mink with the same models for oxygen. In [34], the Moser model was added to the nine models studied. Additional criteria such as the Fisher function and the correlation coefficient were added to the model evaluation criteria. The results obtained after the application of ICDA have shown that the models were reduced at the growth rate from lactose to only three—Mink, Tessier, and Haldane. For the growth rate from oxygen, there were only two—Mink and Haldane. The application of the ICDA has permitted us to investigate only the combination of the models Mink, Tessier, and Haldane for lactose, and Mink and Haldane for oxygen.

We have discrepancies in the preferred models considering the presented results for model selection when applying the ICDA method. This is due to the fact that the research was conducted under different conditions. In the first case [32,33], we have nine models

and eight comparison criteria: criteria of evaluation of the model parameters, statistic λ, correlation coefficient, and Fisher coefficient. In the second case, we have ten models and eleven comparison criteria. A relative error has been added to the above criteria for comparing models. In both cases, the model of yeast *Kluyveromyces marxianus var. lactis* MC 5 is composed of three kinetic variables—concentration of biomass, lactose, and oxygen. The models also include an important indicator of mass transfer—gas hold-up.

In the present work, we aim to make a comparison between the ten growth rate models—Monod, Mink, Tessier, Moser, Aiba, Andrews, Haldane, Luong, Edward, and Han-Levenspiel. The models are developed separately for the two main substrates—lactose and oxygen with perfect mixing in the bioreactor. Mass-transfer and gas hold-up are not reported. The selection of the most suitable models will be made through the application of ICDA. Then, a simulation of the general model of a batch process in the cultivation of *Kluyveromyces marxianus var. lactis* MC 5 will be made. So far, no additive form of specific growth rate has been used in any study of this process. At the end of the manuscript, multiplicative and additive forms of specific growth rates will be compared.

## 2. Materials and Methods

### 2.1. Experimental Investigation

The cultivation conditions [13] of the yeast *Kluyveromyces marxianus var. lactis* MC 5 are shown in Table 1.

**Table 1.** Cultivation conditions of yeast *Kluyveromyces marxianus lactis var. lactis* MC 5.

| | |
|---|---|
| 1. | Nutrient medium with a basic component—whey ultra-filtrate with lactose concentration 44 g/L. The ultra-filtrate is derived from whey separated in the production of white cheese and deproteinization by ultra-filtration on LAB 38 DDS with a membrane of the type GR 61 PP under the following conditions: |

| Temperature | T = 40–43 °C; |
|---|---|
| Input pressure | 0.65 MPa; |
| Output pressure | 0.60 MPa. |

| | |
|---|---|
| 2. | The ultra-filtrate is used in the native condition with lactose concentration 44 g/L. The nutrient medium consists of: |

| $(NH)_4HPO$ | 0.6%; |
|---|---|
| Yeast's autolisate | 5%; |
| Yeast's extract | 1%; |
| pH | 5.0–5.2. |

| | |
|---|---|
| 3. | The gas flow rate $Q_G$ is 60 L/L/h up to the 4th hour and 120 L/L/h up to the end of the process under continuous mixing $n = 800\ \text{min}^{-1}$. |
| 4. | Temperature is 29 °C |
| 5. | The changes in the microbiological process (lactose conversion in yeast's cells to protein) are studied during the strain growth: |
| | (a).  Lactose concentration in the fermentation medium, in oxidation, and assimilation of lactose by *Kluyveromyces Marxianus lactis* MC 5 is determined by enzyme methods through UV tests (Boehringer Manheim, Germany, 1983); |
| | (b).  Concentration of cell mass and protein contents are determined on the basis of the nitrogenium contents (Kjeltek system 1028, Denmark); |
| | (c).  Concentration of the dissolved oxygen in the fermentation medium in the process of oxidation and assimilation of lactose is determined by an oxygen sensor. |
| | (d).  The oxygen sensor that is produced by LKB Vertriebs GmbH is used for the measurement of the oxygen concentration in the fermentation middle. |
| 6. | Duration of the cultivation is $t_f$ = 12 h. |

Six fermentations were performed by an aerobic batch cultivation of *Kluyveromyces Marxianus lactis* MC 5. Experimental studies have been performed on a computer-controlled laboratory bioreactor 2L-M with a magnetic drive [19].

### 2.2. Kinetic Models

The conventional model includes a cell concentration in the dependence between concentrations of the basic energetic substrates: lactose and oxygen. The models are synthesized independently for the lactose and the oxygen:

### 2.2.1. Biomass Production from Lactose Only

$$\frac{dX}{dt} = \rho(S)\,X, \tag{1}$$

$$\frac{dS}{dt} = -\frac{\rho(S)}{Y_{X/S}}\,X, \tag{2}$$

where t—process time, h; X—biomass concentration, g/L; S—lactose concentration, g/L; $\rho$(S)—growth rate of biomass from lactose-only, $h^{-1}$; $Y_{X/S}$—yield coefficients of formation of the biomass from lactose, g/g.

### 2.2.2. Biomass Production from Oxygen Only

$$\frac{dX}{dt} = \rho(C)\,X, \tag{3}$$

$$\frac{dC}{dt} = k_l a(C^* - C) - \frac{\rho(C)}{Y_{X/C}}\,X, \tag{4}$$

where C = oxygen concentration, g/L; $\rho$(C) = growth rate of biomass from oxygen-only, $h^{-1}$; $k_l a$ = mass transfer coefficient, $h^{-1}$; $Y_{X/C}$ = yield coefficients of formation of the biomass from oxygen, g/g.

The initial conditions of both models (1)–(4) are [32–34]: X(0) = 0.2 g/L, S(0) = 44 g/L, C(0) = 6.65 $\times$ $10^{-3}$ g/L, and C* = C(0).

### 2.2.3. System Constraints of Both Processes

Nearly all engineering processes have physical constraints. The lactose and oxygen concentration must be positive all the time for the processes, otherwise, an unrealistic solution in the identification problem will be obtained:

$$g_1(t, \mathbf{x}) = -S(t) \le 0 \tag{5}$$

$$g_2(t, \mathbf{x}) = \frac{S(0) - S(t)}{X(t) - X(0)} - \frac{1}{Y_{X/S}} \le 0 \tag{6}$$

In addition, here are the following constraints for stochiometry of the processes:

$$g_3(t, \mathbf{x}) = -C(t) \le 0, \tag{7}$$

$$g_4(t, \mathbf{x}) = \frac{OTR}{X(t) - X(0)} - \frac{1}{Y_{X/C}} \le 0, \tag{8}$$

where $\mathbf{x}$ = vector of estimated parameters, $\mathbf{x} = \mathbf{x}[\mu_m^S, \ldots, Y_{X/S,}]^T$ for model (1)–(2), and $\mathbf{x} = \mathbf{x}[\mu_m^C, \ldots, Y_{X/C}, k_l a]^T$ for model (3)–(4).

### 2.2.4. Specific Growth Rate Models

This work compares ten unstructured models [35]: $M_1$ = Monod, $M_2$ = Mink, $M_3$ = Tessier, $M_4$ = Moser, $M_5$ = Aiba, $M_6$ = Andrews, $M_7$ = Haldane, $M_8$ = Luong, $M_9$ = Edward, and $M_{10}$ = Han-Levenspiel in order to explain the kinetics of the cell growth.

### 2.2.5. Criteria for Evaluation of the Model Parameters

The least-squares error is commonly employed as a criterion to inspect how close the computed profiles of the state variables come to the experimental observations [36]:

$$Q_1 = \frac{1}{N}\sum_{i=1}^{N}\left(\frac{(X_e(t_i) - X_m(t_i))^2}{X_{emax}^2} + \frac{(S_e(t_i) - S_m(t_i))^2}{S_{emax}^2}\right), \tag{9}$$

$$Q_2 = \frac{1}{N} \sum_{i=1}^{N} \left( \frac{(X_e(t_i) - X_m(t_i))^2}{X_{emax}^2} + \frac{(C_e(t_i) - C_m(t_i))^2}{C_{emax}^2} \right), \tag{10}$$

where $Q_1$ and $Q_2$ = sums of the squares of weighted residuals only from lactose or oxygen; $N$ = number of the experiments, $N$ = 12; $t_i$ = time partitions, h; $X_e(t_i)$, $S_e(t_i)$, $C_e(t_i)$ and $X_m(t_i)$, $S_m(t_i)$, $C_m(t_i)$ = experimental and simulated values of kinetics variables, respectively, g/L; $X_{emax}^2$, $S_{emax}^2$, $C_{emax}^2$ = maximal values of every kinetic variable, g/L.

The constraints are included in the parameter estimation problem (9)–(10), in order to avoid unrealistic predicted values:

$$C_1(S) = \min_{\mathbf{x}} \left( Q_1 + \sum_{i=1}^{2} w_i \int_{t=0}^{t_f} g_i(t, \mathbf{x}) \, dt \right), \tag{11}$$

$$C_1(C) = \min_{\mathbf{x}} \left( Q_2 + \sum_{i=3}^{4} w_i \int_{t=0}^{t_f} g_i(t, \mathbf{x}) \, dt \right), \tag{12}$$

where $\mathbf{x}$ = vector of the estimated parameters; $C_1(S)$ and $C_1(C)$ = criteria of minimization for lactose-only and for oxygen-only; $t_f$ = final time, h; $g_i(t, \mathbf{x})$—system constraints, i = 1, ..., 4; $w_i$ = the weight of each constraints, $w_i = 10^2$. Such a large weight of each constrain is chosen to make it easier to identify models. If the constraints are satisfied then $g_i(t, \mathbf{x}) = 0$.

2.2.6. Criteria for Model Validations

The criteria for evaluating the parameters in the models have been used to validate and verify the adequacy of the models. Statistical criteria: relative error ($S_L$) that shows the error between the experimental and simulated results and the correlation coefficient ($R^2$), which is one of the most commonly used to assess the adequacy of the regression models.

Two criteria have been used to test the null hypothesis: statistics $\lambda$ is the criteria for testing the hypothesis of zero mean deviation of the model and experimental data. This criterion has been used in Giridhar and Srivastava [37]. Bard is cited in this publication, who recommends that this criterion be used to assess the reliability of biotechnological process models. In addition to this criterion for testing the null hypothesis, the standard Fisher coefficient is included.

The formulas for calculating statistic $\lambda$, relative error ($S_L$), Fisher coefficient (F), and correlation coefficient ($R^2$) are fully presented in [34].

The criteria for model validation have the following designations:

- $C_1(S)$ or $C_1(C)$—criteria for evaluation of model parameters (11)–(12);
- $C_2(S)$ or $C_2(C)$—statistics $\lambda$. The criterion $C_2$ was compared to the tabular Fisher coefficient ($F_T^\lambda$) with a degree of freedom $F_T^\lambda(2, 11)$;
- Relative error ($S_L$) for kinetics variables: $C_3'(X) = S_L(X)$ and $C_4(S) = S_L(S)$ for lactose only or $C_3''(X) = S_L(X)$ and $C_4(C) = S_L(C)$ for oxygen only;
- Fisher coefficient (F) for the kinetics variables: $C_5'(X) = F(X)$ and $C_6(S) = F(S)$ for lactose only or $C_5''(X) = F_L(X)$ and $C_6(C) = F(C)$ for oxygen only. Similarly, the obtained values of $C_5'(X)$, $C_6(S)$, $C_5''(X)$, and $C_6(C)$ have been compared with the tabular Fischer coefficient, but for degrees of freedom $F_T(11, 2)$;
- Experimental correlation coefficient ($R^2$) for kinetics variables: $C_7'(X) = R^2(X)$ and $C_8(S) = R^2(S)$ for lactose only or $C_7''(X) = R^2(X)$ and $C_8''(X) = R^2(C)$ and $C_8(C) = R_C^2$. The obtained values of $C_7'(X)$, $C_8(S)$, $C_7''(X)$, and $C_8(C)$ have been compared to the tabular correlation coefficient with a degree of freedom $R_T^2(11)$.

2.3. *Software for Identification of Kinetics Parameters and Computing Criteria*

An algorithm and program of COMPAQ Visual FORTRAN 90 [38] have been developed in order to identify the parameters in the models (1)–(10) for the batch cultivation process of yeast Kluyweromyces marxianus var. lactis MC 5. We have used the average

values of the kinetic variables of the six fermentations for modeling the process. For solving the nonlinear problem (11)–(12), we have used a direct search method, a subroutine BCPOL with a double-precision from IMSL Math/Library [39]. All computations have been performed using HexaCore AMD Phenom II X6 1075T, 3 GHz, 8 GB RAM, Windows XP operating system (32 bit).

*2.4. Background of the ICDA Method*

Following [15–18], from an Intuitionistic Fuzzy Pair (IFP) [22] we have obtained an estimation of the degrees of "agreement" and "disagreement" between two criteria applied to different objects. An IFP is an ordered pair of real non-negative numbers $\langle \alpha, \beta \rangle$ such that $\alpha + \beta \leq 1$.

Consider an IM [19–21] whose index sets consist of the criteria (for rows) and objects (for columns). The elements of this IM are further assumed to be real numbers. An IM with index sets consisting of the criteria (for rows and for columns) with elements IFPs corresponding to the degrees of "agreement" and "disagreement" between the respective criteria is then constructed.

Let O denote the set of all objects $O_1, O_2, \ldots, O_n$ being evaluated, and $C(O)$ be the set of values assigned to the objects by a given criterion C, i.e.,

$$O \stackrel{\text{def}}{=} \{O_1, O_2, \ldots, O_n\}; \; C(O) \stackrel{\text{def}}{=} \{C(O_1), C(O_2), \ldots, C(O_n)\}.$$

Then, let $C^*(O) \stackrel{\text{def}}{=} \{\langle x, y \rangle | \; x \neq y \; \& \; \langle x, y \rangle \in C(O) \times C(O)\}$.

In order to find the "agreement" between two criteria, the vector of all internal comparisons of each criterion, which fulfills exactly one of the following three relations: R, $\overline{R}$, and $\widetilde{R}$, is constructed. In other words, it is required that for a fixed criterion C and any ordered pair $\langle x, y \rangle \in C^*(O)$ it is true that:

$$\langle x, y \rangle \in R \Leftrightarrow \langle y, x \rangle \in \overline{R}, \tag{13}$$

$$\langle x, y \rangle \in \widetilde{R} \Leftrightarrow \langle x, y \rangle \notin (R \cup \overline{R}), \tag{14}$$

$$R \cup \overline{R} \cup \widetilde{R} = C^*(O) \tag{15}$$

Only a subset of $C(O) \times C(O)$ needs to be considered for the effective calculation of the vector of internal comparisons (further denoted by $V(C)$) since from (13)–(15) it follows that if the relation between x and y is known, then so is the relation between y and x. Thus, of interest are only the lexicographically ordered pairs $\langle x, y \rangle$. Denote for brevity $C_{i,j} = \langle C(O_i), C(O_j) \rangle$. Then, for a fixed criterion C, the vector with $n(n-1)/2$ elements is obtained:

$$V(C) = \{C_{1,2}, C_{1,3}, \ldots, C_{1,n}, C_{2,3}, C_{2,4}, \ldots, C_{2,n}, C_{3,4}, \ldots, C_{3,n}, \ldots, C_{n-1,n}\}$$

Let $V(C)$ be replaced by $\hat{V}(C)$, where for the k-th component $(1 \leq k \leq \frac{n(n-1)}{2})$:

$$\hat{V}_k(C) = \begin{cases} 1, & \text{iff } V_k(C) \in R \\ -1, & \text{iff } V_k(C) \in \overline{R} \\ 0, & \text{otherwise} \end{cases}$$

When comparing two criteria, the degree of "agreement" $\mu_{C,C'}$ is determined as the number of matching components of the respective vectors, divided by the length of the vector for normalization purposes. The degree of "disagreement" $\nu_{C,C'}$ is the number of components of opposing signs in the two vectors, again normalized by the length. The ordered pair $\langle \mu_{C,C'}, \nu_{C,C'} \rangle$ is an IFP. In most of the obtained pairs, the sum $\mu_{C,C'} + \nu_{C,C'}$ is equal to 1. However, there may be some pairs, for which this sum is less than 1. The difference $\pi_{C,C'} = 1 - \mu_{C,C'} - \nu_{C,C'}$ is considered as a degree of "uncertainty".

Let $\alpha, \beta \in [0, 1]$ be given, so that $\alpha + \beta \leq 1$. The criteria C and C' are in:

- $\langle \alpha, \beta \rangle$—Positive consonance (PC), if $(\mu_{C,C'} > \alpha)$ and $(\nu_{C,C'} < \beta)$;
- $\langle \alpha, \beta \rangle$—Negative consonance (NC), if $(\mu_{C,C'} < \beta)$ and $(\nu_{C,C'} > \alpha)$;
- $\langle \alpha, \beta \rangle$—Dissonance (D), otherwise.

The method is described in detail in the publications of colleagues [14,40–42].

Research implies a more detailed description of the membership function and non-membership function (Table 2).

**Table 2.** Intuitionistic fuzzy threshold values for membership and non-membership function.

| Level for Positive Consonance | Membership Function | Non-Membership Function |
|---|---|---|
| Weak PC | $0.75 \leq \mu < 0.85$ | $0.15 \leq \nu < 0.25$ |
| Good PC | $0.85 \leq \mu < 0.90$ | $0.10 \leq \nu < 0.15$ |
| Very good PC | $0.90 \leq \mu < 0.95$ | $0.05 \leq \nu < 0.10$ |
| Strong PC | $0.95 \leq \mu < 0.98$ | $0.02 \leq \nu < 0.05$ |
| Very Strong PC | $0.98 \leq \mu < 1.00$ | $0.00 \leq \nu < 0.02$ |
| Strict PC | $\mu = 1.00$ | $\nu = 0.00$ |

## 3. Results and Discussions

### 3.1. Results for Independent Process Models

Tables 3 and 4 show the results for the criteria by which the models ($M_1$–$M_{10}$) are validated from lactose only (1)–(2) and from oxygen only (3)–(4).

**Table 3.** Criteria for the model validation from lactose only.

| Models | $C_1(S) \times 10^{-3}$ | $C_2(S)$ | $C_{3'}(X)$ | $C_4(S)$ | $C_{5'}(X)$ | $C_6(S)$ | $C_{7'}(X)$ | $C_8(S)$ |
|---|---|---|---|---|---|---|---|---|
| $M_1(S)$ | 13.8827 | 151.8377 | 0.2262 | 0.5030 | 1.0125 | 1.1676 | 0.9864 | 0.9754 |
| $M_2(S)$ | 9.6736 | 204.1643 | 0.2058 | 0.6288 | 1.0598 | 1.0253 | 0.9744 | 0.9920 |
| $M_3(S)$ | 10.6736 | 151.3657 | 0.1982 | 0.3879 | 1.0220 | 1.1364 | 0.9869 | 0.9819 |
| $M_4(S)$ | 9.2521 | 175.7246 | 0.1960 | 0.3883 | 1.0471 | 1.0591 | 0.9790 | 0.9894 |
| $M_5(S)$ | 14.0315 | 151.7896 | 0.2274 | 0.5067 | 1.0124 | 1.1690 | 0.9863 | 0.9752 |
| $M_6(S)$ | 13.9697 | 151.9010 | 0.2270 | 0.5051 | 1.0129 | 1.1684 | 0.9863 | 0.9753 |
| $M_7(S)$ | 13.9765 | 152.1378 | 0.2272 | 0.5045 | 1.0130 | 1.1681 | 0.9864 | 0.9752 |
| $M_8(S)$ | 13.8925 | 151.7663 | 0.2262 | 0.5036 | 1.0128 | 1.1679 | 0.9864 | 0.9754 |
| $M_9(S)$ | 14.1648 | 151.9544 | 0.2289 | 0.5087 | 1.0118 | 1.1698 | 0.9863 | 0.9749 |
| $M_{10}(S)$ | 6.0574 | 184.7091 | 0.1641 | 0.1845 | 1.0694 | 1.0313 | 0.9853 | 0.9944 |

**Table 4.** Criteria for the model validation from oxygen only.

| Models | $C_1(C) \times 10^{-3}$ | $C_2(C)$ | $C_{3''}(X)$ | $C_4(C)$ | $C_{5''}(X)$ | $C_6(C)$ | $C_{7''}(X)$ | $C_8(C)$ |
|---|---|---|---|---|---|---|---|---|
| $M_1(C)$ | 0.6734 | 154.5335 | 0.1612 | 0.5992 | 1.0127 | 1.0174 | 0.9992 | 0.9986 |
| $M_2(C)$ | 1.5593 | 163.0421 | 0.1864 | 0.9618 | 1.0187 | 1.0155 | 0.9988 | 0.9963 |
| $M_3(C)$ | 1.0059 | 161.9099 | 0.1748 | 0.8047 | 1.0141 | 1.0203 | 0.9991 | 0.9978 |
| $M_4(C)$ | 0.6727 | 156.9493 | 0.1596 | 0.6473 | 1.0115 | 1.0223 | 0.9992 | 0.9988 |
| $M_5(C)$ | 1.1461 | 160.2555 | 0.1818 | 0.2635 | 1.0174 | 1.0178 | 0.9989 | 0.9975 |
| $M_6(C)$ | 1.4423 | 172.3611 | 0.2021 | 0.2539 | 1.0223 | 1.0235 | 0.9989 | 0.9969 |
| $M_7(C)$ | 1.6108 | 169.1571 | 0.1941 | 0.2713 | 1.0195 | 1.0286 | 0.9989 | 0.9965 |
| $M_8(C)$ | 1.1806 | 165.3167 | 0.1869 | 0.2492 | 1.0218 | 1.0164 | 0.9989 | 0.9975 |
| $M_9(C)$ | 1.1256 | 159.8748 | 0.1812 | 0.2646 | 1.0176 | 1.0169 | 0.9989 | 0.9975 |
| $M_{10}(C)$ | 1.0604 | 163.3446 | 0.1855 | 0.2611 | 1.0226 | 1.0111 | 0.9990 | 0.9978 |

The minimal ($C_{imin}$) and maximal ($C_{imax}$) values of every criterion for lactose only and oxygen only are shown in Table 5.

**Table 5.** Intervals of change of lactose only and oxygen only criteria.

| | Lactose Only | | | Oxygen Only | |
|---|---|---|---|---|---|
| **Criteria** | $\mathbf{C_{imin} \times 10^{-3}}$ | $\mathbf{C_{imax} \times 10^{-3}}$ | **Criteria** | $\mathbf{C_{imin} \times 10^{-3}}$ | $\mathbf{C_{imax} \times 10^{-3}}$ |
| $C_1(S)$ | 6.0574 | 14.1648 | $C_1(C)$ | 0.6727 | 1.6108 |
| $C_2(S)$ | 151.3657 | 204.1643 | $C_2(C)$ | 154.5335 | 172.3611 |
| $C_{3'}(X)$ | 0.1641 | 0.2289 | $C_3''(X)$ | 0.1596 | 0.2021 |
| $C_4(S)$ | 0.1845 | 0.6288 | $C_4(C)$ | 0.2492 | 0.9618 |
| $C_{5'}(X)$ | 1.0118 | 1.0694 | $C_5''(X)$ | 1.0115 | 1.0226 |
| $C_6(S)$ | 1.0253 | 1.1698 | $C_6(C)$ | 1.0111 | 1.0286 |
| $C_{7'}(X)$ | 0.9744 | 0.9869 | $C_7''(X)$ | 0.9988 | 0.9992 |
| $C_8(S)$ | 0.9749 | 0.9944 | $C_8(C)$ | 0.9963 | 0.9988 |

The criteria $C_2$, $C_5$, and $C_6$ are the criteria for testing the zero hypothesis. These criteria have tabular values to which they are compared to. The tabular values of $C_2$ and $C_5$–$C_8$ are given from statistical tables [43]. Fisher coefficient for $C_2$ (statistic $\lambda$) is $F_T^\lambda(2, 11) = 4.04$. For Fisher coefficients ($C_5$ and $C_6$) it is $F_T(11, 2) = 19.40$, and for correlation coefficients ($C_7$ and $C_8$) the tabular value is $R_T^2(11) = 0.684$, for level of significance $\alpha = 0.01$. The criteria $C_2 > F_T^\lambda(2, 11)$, the experimental Fisher coefficients, criteria ($C_5$ and $C_6$) < $F_T(11, 2)$, and the experimental correlation coefficients criteria ($C_7$ and $C_8$) > $R_T^2(11) = 0.684$. The presented results show that in terms of the criteria for validation ($C_2$, $C_5$–$C_8$) all growth rate models depending on lactose only or oxygen only are adequate.

Figure 1 shows the results for the biomass formation depending on lactose only or oxygen only.

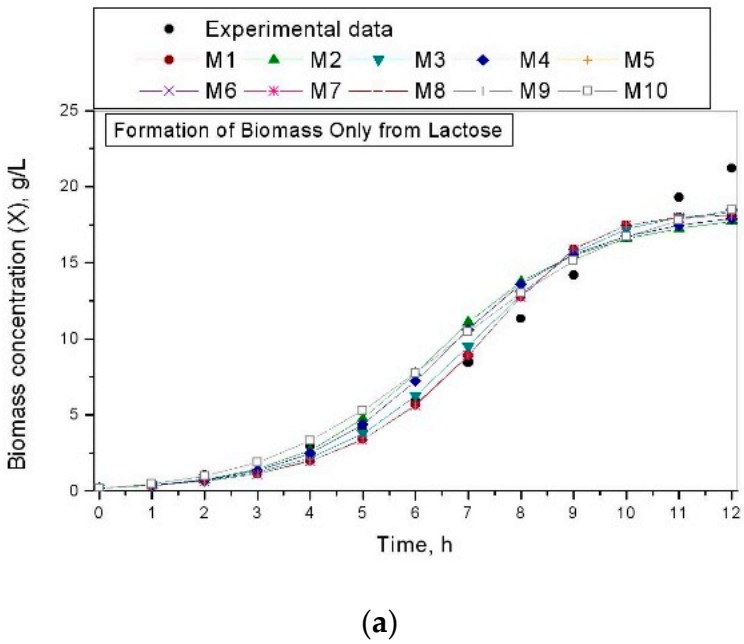

(**a**)

**Figure 1.** *Cont.*

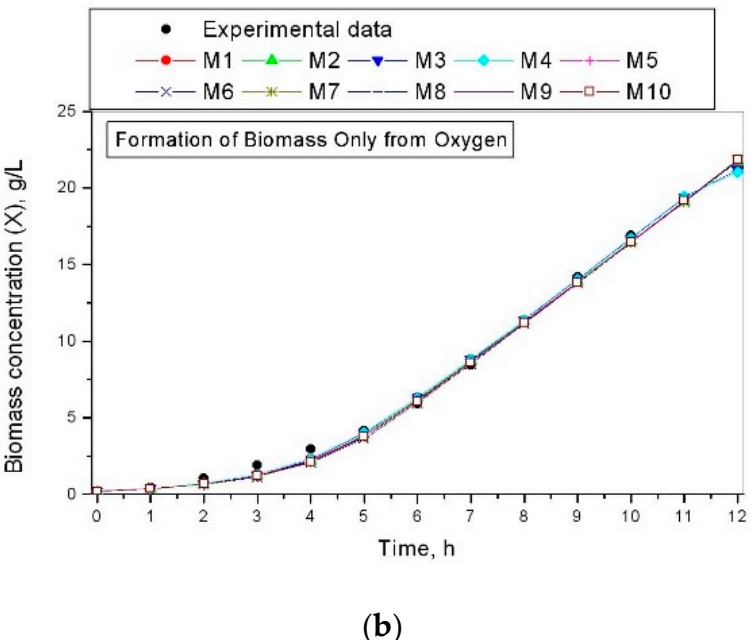

(**b**)

**Figure 1.** Formation of biomass from investigated models: (**a**) formation of biomass only from lactose; (**b**) formation of biomass only from oxygen.

Figure 1a for the biomass concentration dependent on lactose-only presents that there is a discrepancy between the experimental and simulated data for the kinetic variable.

Figure 1b for the biomass concentration dependent on oxygen-only presents that there is a different situation. All investigated growth rate models describe very well the experimental values of the kinetic variables of the process.

Figure 2 shows the results of lactose or oxygen concentration for models (1)–(2) and (3)–(4).

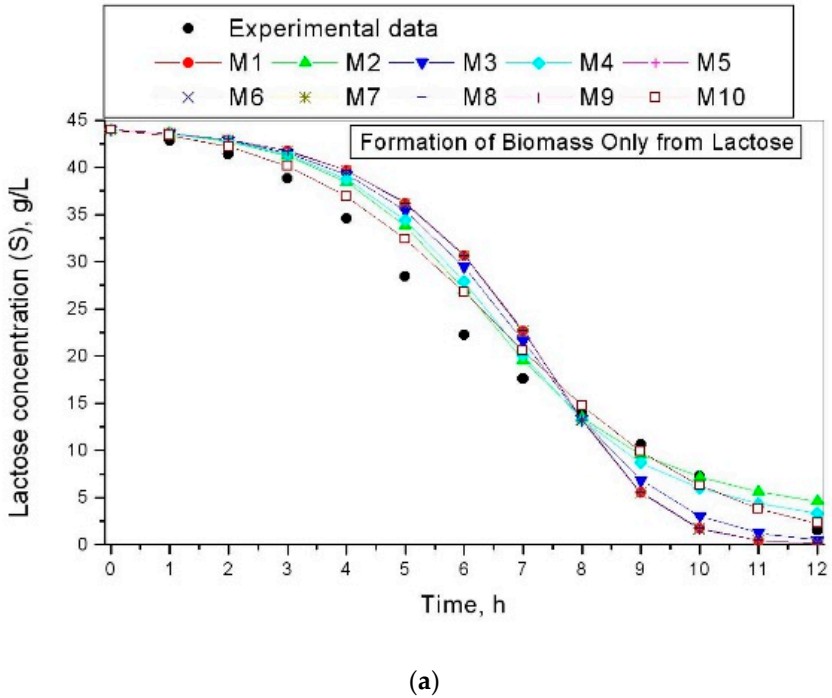

(**a**)

**Figure 2.** *Cont.*

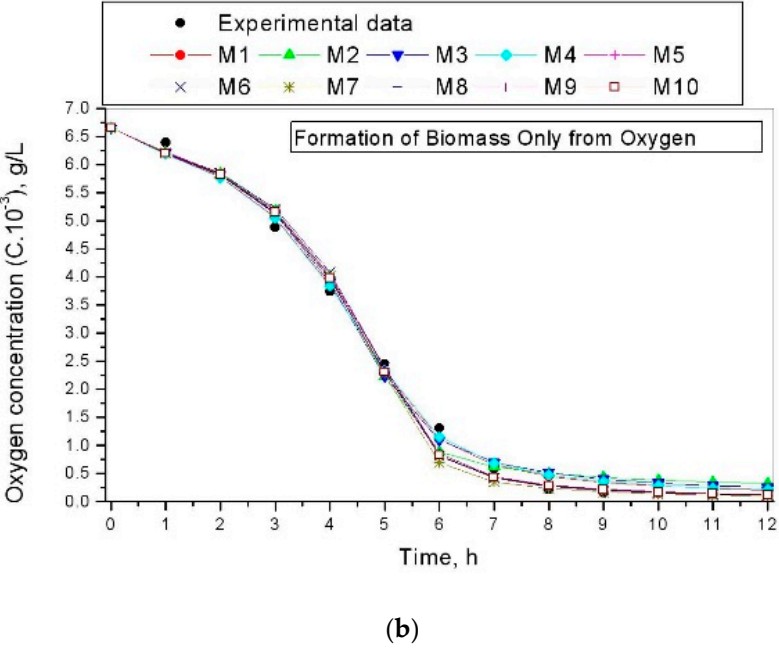

(**b**)

**Figure 2.** Lactose and oxygen concentration for investigated models: (**a**) lactose concentration for models (1)–(2); (**b**) oxygen concentration for models (3)–(4).

Figure 2a,b shows a similar situation to the reasoning made for Figure 1a,b. The modeling of the lactose concentration model (1)–(2) shows a relatively lower correspondence between the experimental and simulated results compared to the results obtained by models (3)–(4), shown in Figure 2b. These results show that the influence of oxygen on the cell growth kinetics is more than that of lactose.

The application of the ICDA method will show which models will drop out.

### 3.2. Application of ICDA Method

In ICDA, the method is included in the criteria $C_1$–$C_8$. The objects (alternative) in the ICDA are the investigated growth rate models. The method is used for the selection of the growth rate for independent models (1)–(2) and (3)–(4). For selecting growth rate models, we have to have many high values of the membership function ($\mu$) because we have accepted $\alpha = 0.95$ and $\beta = 0.05$.

The values of the membership function ($\mu$) and non-membership function ($\nu$) have been calculated with the help of the software (ICrAData) developed by our colleagues [44] for the realization of the method. ICrAData applies five different algorithms for calculating inter-criteria relationships, namely: $\mu$-biased, Unbiased, $\nu$-biased, Balanced, and Weighted. The ICrAData software displays results in two panels—matrix and graphical. The results can be exported in different formats: matrices, vectors, and graphs. In the matrix view, the data from the columns can be sorted in ascending or descending order. The graphical view has options for resizing the intuitionistic fuzzy triangle, displaying a grid, and assigning different colors to the dots.

The indexed matrix ($\mathbf{B}_1{}^{\mu}$) with membership function ($\mu$) for lactose only is shown in Table 6.

**Table 6.** Index matrix $\mathbf{B}_1{}^\mu$ for lactose only.

| $\mathbf{B}_1{}^\mu$ = | $M_1(S)$ | $M_2(S)$ | $M_3(S)$ | $M_4(S)$ | $M_5(S)$ | $M_6(S)$ | $M_7(S)$ | $M_8(S)$ | $M_9(S)$ | $M_{10}(S)$ |
|---|---|---|---|---|---|---|---|---|---|---|
| $M_1(S)$ | 1 | 0.9286 | 1.0000 | 0.9643 | 1.0000 | 1.0000 | 1.0000 | 1.0000 | 1.0000 | 0.9286 |
| $M_2(S)$ | 0.9286 | 1 | 0.9286 | 0.9643 | 0.9286 | 0.9286 | 0.9286 | 0.9286 | 0.9286 | 1.0000 |
| $M_3(S)$ | 1.0000 | 0.9286 | 1 | 0.9643 | 1.0000 | 1.0000 | 1.0000 | 1.0000 | 1.0000 | 0.9286 |
| $M_4(S)$ | 0.9643 | 0.9643 | 0.9643 | 1 | 0.9643 | 0.9643 | 0.9643 | 0.9643 | 0.9643 | 0.9643 |
| $M_5(S)$ | 1.0000 | 0.9286 | 1.0000 | 0.9643 | 1 | 1.0000 | 1.0000 | 1.0000 | 1.0000 | 0.9286 |
| $M_6(S)$ | 1.0000 | 0.9286 | 1.0000 | 0.9643 | 1.0000 | 1 | 1.0000 | 1.0000 | 1.0000 | 0.9286 |
| $M_7(S)$ | 1.0000 | 0.9286 | 1.0000 | 0.9643 | 1.0000 | 1.0000 | 1 | 1.0000 | 1.0000 | 0.9286 |
| $M_8(S)$ | 1.0000 | 0.9286 | 1.0000 | 0.9643 | 1.0000 | 1.0000 | 1.0000 | 1 | 1.0000 | 0.9286 |
| $M_9(S)$ | 1.0000 | 0.9286 | 1.0000 | 0.9643 | 1.0000 | 1.0000 | 1.0000 | 1.0000 | 1 | 0.9286 |
| $M_{10}(S)$ | 0.9286 | 1.0000 | 0.9286 | 0.9643 | 0.9286 | 0.9286 | 0.9286 | 0.9286 | 0.9286 | 1 |

We are not interested in models that have dissonance. From Table 6 it can be seen that we have many models that have a strict PC ($\mu$ = 1.00) and a strong PC (Table 1).

The strict PC is obtained by models: $M_1(S) \leftrightarrow M_3(S) \leftrightarrow M_5(S) \leftrightarrow M_6(S) \leftrightarrow M_7(S) \leftrightarrow M_8(S) \leftrightarrow M_9(S)$; $M_2(S) \leftrightarrow M_{10}(S)$, and the strong PC—by models:

$M_1(S) \leftrightarrow M_2(S) \leftrightarrow M_3(S) \leftrightarrow M_4(S) \leftrightarrow M_5(S) \leftrightarrow M_6(S) \leftrightarrow M_7(S) \leftrightarrow M_8(S) \leftrightarrow M_9(S) \leftrightarrow M_{10}(S)$.

After applying the ICDA method for analysis and evaluation of growth rate models depending on lactose only, it turns out that the most suitable model is the $M_1(S)$ Monod model.

The indexed matrix ($\mathbf{B}_2{}^\mu$) with membership function ($\mu$) for oxygen only is shown in Table 7.

**Table 7.** Index matrix $\mathbf{B}_2{}^\mu$ for oxygen only.

| $\mathbf{B}_2{}^\mu$ = | $M_1(C)$ | $M_2(C)$ | $M_3(C)$ | $M_4(C)$ | $M_5(C)$ | $M_6(C)$ | $M_7(C)$ | $M_8(C)$ | $M_9(C)$ | $M_{10}(C)$ |
|---|---|---|---|---|---|---|---|---|---|---|
| $M_1(C)$ | 1 | 0.8214 | 0.9286 | 1.0000 | 0.8571 | 0.8571 | 0.8571 | 0.8214 | 0.8214 | 0.8214 |
| $M_2(C)$ | 0.8214 | 1 | 0.8929 | 0.8214 | 0.9643 | 0.9643 | 0.9643 | 1.0000 | 1.0000 | 1.0000 |
| $M_3(C)$ | 0.9286 | 0.8929 | 1 | 0.9286 | 0.9286 | 0.9286 | 0.9286 | 0.8929 | 0.8929 | 0.8929 |
| $M_4(C)$ | 1.0000 | 0.8214 | 0.9286 | 1 | 0.8571 | 0.8571 | 0.8571 | 0.8214 | 0.8214 | 0.8214 |
| $M_5(C)$ | 0.8571 | 0.9643 | 0.9286 | 0.8571 | 1 | 1.0000 | 1.0000 | 0.9643 | 0.9643 | 0.9643 |
| $M_6(C)$ | 0.8571 | 0.9643 | 0.9286 | 0.8571 | 1.0000 | 1 | 1.0000 | 0.9643 | 0.9643 | 0.9643 |
| $M_7(C)$ | 0.8571 | 0.9643 | 0.9286 | 0.8571 | 1.0000 | 1.0000 | 1 | 0.9643 | 0.9643 | 0.9643 |
| $M_8(C)$ | 0.8214 | 1.0000 | 0.8929 | 0.8214 | 0.9643 | 0.9643 | 0.9643 | 1 | 1.0000 | 1.0000 |
| $M_9(C)$ | 0.8214 | 1.0000 | 0.8929 | 0.8214 | 0.9643 | 0.9643 | 0.9643 | 1.0000 | 1 | 1 |
| $M_{10}(C)$ | 0.8214 | 1.0000 | 0.8929 | 0.8214 | 0.9643 | 0.9643 | 0.9643 | 1.0000 | 1.0000 | 1 |

We are not interested in models that have dissonance (Table 7). From Table 7, it can be seen that we have many models that have a strict PC ($\mu$ = 1.00) and a strong PC (Table 1).

A strict PC and a strong PC is possessed by the models: $M_1(C) \leftrightarrow M_4(C)$ and $M_2(C) \leftrightarrow M_5(C) \leftrightarrow M_6(C) \leftrightarrow M_7(C) \leftrightarrow M_8(C) \leftrightarrow M_9(C) \leftrightarrow M_{10}(C)$.

The analysis of the results (Table 7) shows that for oxygen only, we have two models with a strict PC—$M_1(C)$ and with a strong PC—$M_2(C)$.

The results obtained after the application of ICDA (Tables 6 and 7) show that the most suitable model for the growth rate for lactose only is the Monod model, and for oxygen only we have two models—Monod and Mink.

Therefore, for a specific growth rate of the process we have two cases:

Case 1:

$$\mu_1(S, C) = \frac{\mu_{\max}(1)\,S}{S + K_S(1)} \frac{C}{K_C(1) + C} \tag{16}$$

Case 2:

$$\mu_2(S, C) = \frac{\mu_{\max}(2)\,S}{S + K_S(2)} \frac{C^2}{K_C(2) + C^2} \tag{17}$$

The kinetic model of the process for both cases has the form:

$$\frac{dX}{dt} = \mu_j(S, C) \, X, \frac{dS}{dt} = -\frac{1}{Y_{X/S}(j)} \frac{dX}{dt}, \frac{dC}{dt} = -\frac{1}{Y_{X/C}(j)} \frac{dX}{dt} + k_l a(j)(C^* - C), \quad (18)$$

where j = 1—Case 1, and j = 2—Case 2.

For the first time, the additive form of a specific growth rate will be studied to model this process:

Case 3:

$$\mu_3(S) = \frac{\mu_{max}(3) \, S}{S + K_S(3)}, \text{ and } \mu_4(C) = \frac{\mu_{max}(4)C}{K_C(4) + C} \quad (19)$$

Case 4:

$$\mu_5(S) = \frac{\mu_{max}(5) \, S}{S + K_S(5)}, \text{ and } \mu_6(C) = \frac{\mu_{max}(6)C^2}{K_C(6) + C^2} \quad (20)$$

The formation of biomass over time from the two main substrates, lactose and oxygen, has the form:

$$\frac{dX}{dt} = [\mu_k(S) + \mu_k(C)] \, X \quad (21)$$

The concentration of lactose over time depends only on the growth rate of lactose:

$$\frac{dS}{dt} = -\frac{1}{Y_{X/S}(j)} \mu_k(S) \, X \quad (22)$$

The concentration of oxygen over time depends only on the growth rate of oxygen:

$$\frac{dC}{dt} = -\frac{1}{Y_{X/C}(j)} \mu_k(C) \, X + k_l a(k)(C^* - C) \quad (23)$$

where for Case 3 → j = 3, and k = 3, 4, and for Case 4 → j = 4, and k = 5, 6.

The next part of the work will present the modeling and comparison between the considered variants of specific growth rates.

### 3.3. Results after Modeling of Processes for Four Different forms for the Specific Growth Rate

Another algorithm and program of COMPAQ Visual FORTRAN 90 [38] have been developed in order to identify the parameters in models (13)–(20). All computations have been performed using HexaCore AMD Phenom II X6 1075T, 3 GHz, 8 GB RAM, Windows XP operating system (32 bit).

Table 8 shows the calculated values of the kinetic parameters, after identification of models (16)–(23).

**Table 8.** Values of parameters in models, where i = 1, 4, and *j* = 1, 6.

| Case | $\mu_{max}(j)$ | $\mu_{max}(j)$ | $K_S(i)$ | $K_C(i)$ | $k_l a(i)$ | $Y_{X/S}(i)$ | $Y_{X/C}(i)$ |
|------|------|------|------|------|------|------|------|
| 1 | 0.7148 | – | 0.0499 | $0.5034 \times 10^{-3}$ | 131.8461 | 0.4346 | 4.8992 |
| 2 | 0.6914 | – | 0.0116 | $0.3327 \times 10^{-6}$ | 164.6461 | 0.4347 | 3.9980 |
| 3 | 0.3485 | 0.5313 | 44.9874 | $0.5796 \times 10^{-3}$ | 132.8683 | 0.1000 | 3.5230 |
| 4 | 0.2343 | 0.5027 | 14.7631 | $0.1005 \times 10^{-6}$ | 163.5107 | 0.1036 | 3.7581 |

The simulation results for biomass, lactose, and oxygen concentration for the investigated model of specific growth rates are shown in Figure 3.

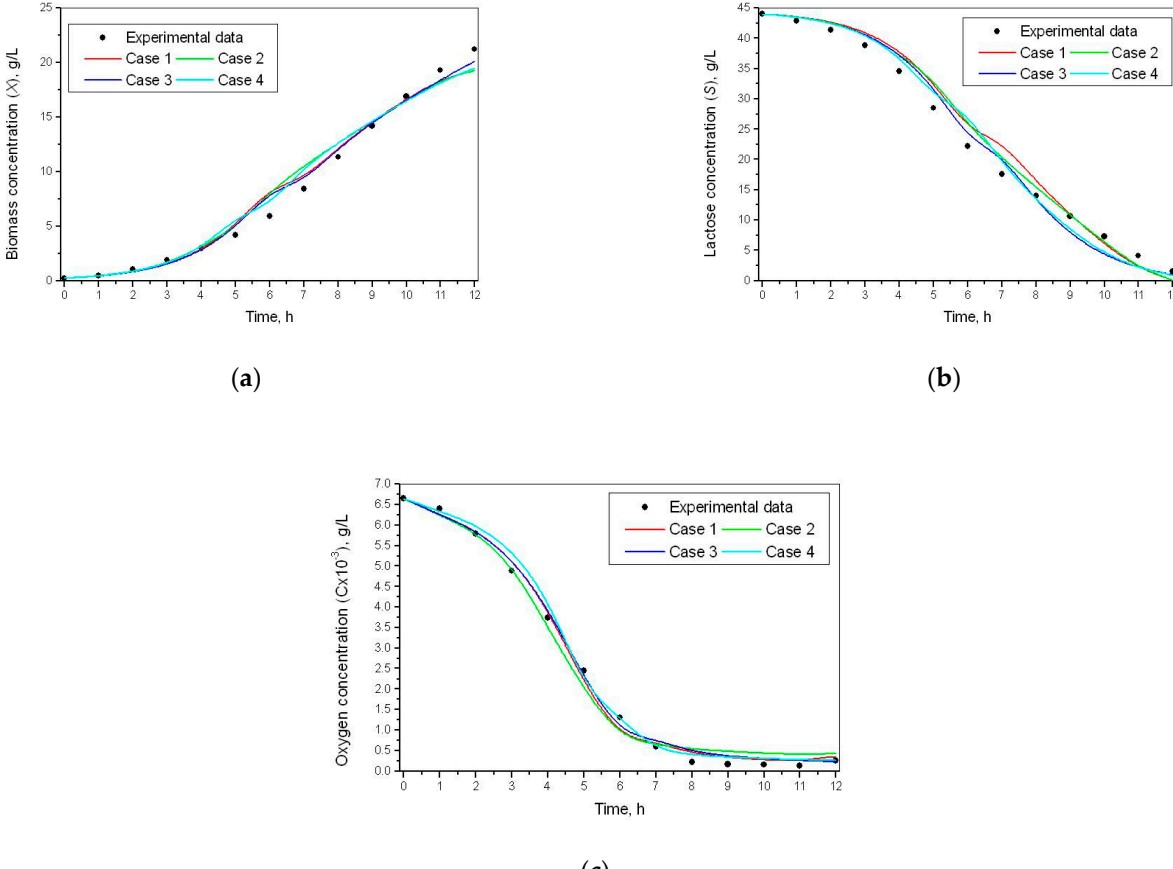

**Figure 3.** Simulation results for kinetics variables of the process. (**a**) Biomass concentration; (**b**) lactose concentration; (**c**) oxygen concentration.

The criteria for comparing the models (16)–(23) are different from the criteria used in models (1)–(4). In this case, instead of eight criteria, we have eleven. This is due to the fact that we already have three kinetic variables of the process—X, S, and C:

- $C_1(j)$ = criteria for evaluation of kinetic parameters in four models of specific growth rate, j = 1, . . . , 4;
- $C_2(j)$ = statistic $\lambda$ for the kinetics variables of four models of specific growth rate, j = 1, . . . , 4;
- $C_3(X)$, $C_4(S)$, $C_5(C)$ = relative error ($S_L$) for the kinetics variables of four models of specific growth rate;
- $C_6(X)$, $C_7(S)$, $C_8(C)$ = experimental Fisher coefficient for the kinetics variables of four models of specific growth rate;
- $C_9(X)$, $C_{10}(S)$, $C_{11}(C)$ = experimental correlation coefficient $R^2$ for kinetics variables of four models of specific growth rate.

Criteria $C_1(j)$ for evaluation of kinetics parameters in models (16)–(23) are determined by the following dependence:

$$C_1(j) = \frac{1}{N}\sum_{i=1}^{N}\left(\frac{(X_e(t_i) - X_m(t_i))^2}{X_{emax}^2} + \frac{(S_e(t_i) - S_m(t_i))^2}{S_{emax}^2} + \frac{(C_e(t_i) - C_m(t_i))^2}{C_{emax}^2}\right) \quad (24)$$

where $C_1(j)$—sums of the squares of weighted residuals for four models, j = 1, . . . , 4.

Table 9 shows the calculated values of the criteria for different cases.

**Table 9.** Values for the criteria by a different case.

| Case | $C_1(j)$ | $C_2(j)$ | $C_3(X)$ | $C_4(S)$ | $C_5(C)$ | $C_6(X)$ | $C_7(S)$ | $C_8(C)$ | $C_9(X)$ | $C_{10}(S)$ | $C_{11}(C)$ |
|------|----------|----------|----------|----------|----------|----------|----------|----------|----------|------------|------------|
| 1 | **7.090** | **150.217** | **0.203** | **0.363** | **0.617** | 1.044 | **1.064** | **1.002** | 0.9904 | **0.9915** | 0.9963 |
| 2 | 7.035 | 148.653 | **0.187** | 0.351 | **1.204** | 1.044 | **1.064** | **1.032** | 0.9889 | **0.9941** | **0.9948** |
| 3 | **5.068** | **137.593** | **0.187** | **0.260** | 0.711 | **1.030** | **1.090** | 1.005 | **0.9936** | 0.9940 | **0.9971** |
| 4 | 6.553 | 143.621 | 0.191 | 0.264 | 0.644 | **1.046** | 1.082 | 1.017 | 0.9907 | 0.9930 | 0.9959 |

Here is the moment to specify that for criteria C1–C8 we are looking for a minimum, and for criteria C9–C11 we are looking for a maximum (Table 9). The better values of criteria C1–C11 are marked in green. The maximum values for criteria C1–C8, and the minimum values for criteria C9–C11 are marked in red.

Again, we apply ICDA to the results shown in Table 9. The results are shown in Table 10.

**Table 10.** Values for the criteria by a different case.

| Case | 1 | 2 | 3 | 4 |
|------|------|--------|------|------|
| 1 | 1 | 0.8909 | 1 | 1 |
| 2 | 0.8909 | 1 | 0.8909 | 0.8909 |
| 3 | 1 | 0.8909 | 1 | 1 |
| 4 | 1 | 0.8909 | 1 | 1 |

Table 10 shows that we have a strict PC for Case 1 ↔ Case 3 ↔ Case 4. These results show that models (16) and (19)–(20) are completely identical. Model (17) is dropped out because $\mu = 0.8909 < 0.95$.

The presented results (Figure 3, Table 9) show that the additive form of Mono, the Monod model (Case 3) has better indicators of the criteria. Case 3 has better results than the other two choices. Criteria $C_1$–$C_4$ and $C_7$ have a minimum, and criteria $C_9$ and $C_{11}$ have a maximum. Therefore, this case has been chosen as a model for the specific growth rate.

## 4. Conclusions

This work has studied ten unstructured models for the growth rate of batch cultivation of yeast *Kluyweromyces marxianus var. lactis* MC 5 depending on the lactose and oxygen, which are the two main substrates of the processes. Two independent kinetic models have been used for the first time in the modeling for lactose and oxygen. The comparison of the results has been achieved by using all criteria. Statistic $\lambda$, Fisher function, and correlation coefficient have shown that all ten models are adequate and can be used for the modeling of the process. The studies made show that there are better results in the kinetic model dependent only on oxygen. This shows that the kinetics of biomass growth is influenced more by oxygen than by lactose.

The application of the ICDA method shows that the most appropriate pair of models is the Monod model for lactose. For oxygen, there are two models—Monod and Mink. The application of the method for the establishment of the correlation connections between the different models for the growth rate for modeling of these processes is very useful not only for assessment and eliminating of equivalence models but also for determination (by dissonance) of how much each of the patterns is closer to the actual experimental data. In establishing correlations between these types of models using ICDA we cannot talk about negative consonance; this would mean that the models do not reflect the real situation of the process and the purpose should have a completely different description of the growth kinetics.

This work also studies for the first time the additive form of a specific growth rate for the modeling of this process, not only the multiplicative form. The idea for studying this form comes from the fact that two independent kinetic models have been used–for lactose and oxygen. The process has been modeled for the four cases of specific growth rates.

Both options of the multiplicative form of the combinations of models Monod–Monod and Monod–Mink have been studied, as well as their two options for their additive forms.

The comparison between the multiplicative and additive forms shows that the additive form of the specific growth rate has better results. The additive form of the Monod–Monod model has been selected for modeling of the specific growth rate.

The ICDA method has one drawback. The method does not take into account what the criterion is, whether a minimum or a maximum is sought. The ICDA has another limitation—the objects and criteria cannot be less than two, i.e., the minimum dimension of the indexed matrix is $2 \times 2$.

In our future work, we will perform a multi-objective parameter estimation for the six fermentations and a sensitivity analysis in a batch cultivation of *Kluyveromyces marxianus var. lactis* MC 5. In this way, the maximum values of the model parameters will be determined, and they can be used for optimization and optimal control of the process.

The approach can be used in the cultivation of stem cells, mammalian cell cultures, and other processes that consume oxygen. The application of the ICDA method in various fields of biotechnology is quite possible.

**Funding:** This research received no external funding.

**Conflicts of Interest:** The author declares no conflict of interest.

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
