# Peer review of "Modeling and Use of Inter-Criteria Decision Analysis for Selecting Growth Rate Models for Batch Cultivation of Yeast Kluyveromyces marxianus var. lactis MC 5"

_fermentation, doi:10.3390/fermentation7030163_

Round 1

Reviewer 1 Report

In this paper, the author reports on the comparison of 10 unstructured models to describe the growth of the yeast Kluyweromyces marxianus var. lactis MC 5 based on lactose and oxygen. Overall, this paper is of interest to the readership of the special issue. I recommend accepting the paper after considering the following points:

Specific comments:

L42: for all applications of the strain, only one reference [2] is included. Please add more references in this part.

L43: The author stated “oxidation of lactose for biomass formation has not been well studied”. Please include a reason to study such an effect for the respective strain.

L44: There are efforts towards generalized models in biotechnology, please add some. E.g., DOI: 10.1007/10_2020_152. 

L48 – L55: A better link of these sentences to the above-mentioned research question is needed. 

Introduction: Please highlight the advantages of such an optimized model and focus on the need to describe the growth based on 2 different substrates. 

L67: Please include the conditions in a table with a table caption.

L69: Please include the conditions in a table with a table caption.

L101: On which basis were the initial conditions defined?

L146: Is such an intensive discussion of the background needed? Isn´t the background explained in suitable references?

L208-214: Please include this part in Materials and Methods

Table 2 and Table 3: What is the ratio of 4 decimal places? In my opinion, 2 would be enough

Figure 2: Please increase size of figure. For the readers, figures are much better to follow the content.

Experimental data for model fitting: What is the ratio to use the same experimental conditions in all experiments? In my opinion, using different experimental conditions would lead to better model comparison.

L233: If both model classes independently describing the growth sufficiently, why would whom combine both model classes? What would be the aim of an optimized model?

For Table 5 and Table 6. Couldn´t you use colors for visualization?

General: I miss a profound discussion of the results and why specific models are more advantageous while others are not?

Conclusion: Could this approach be transferred to different biotechnological areas, e.g., stem cell cultivation, or mammalian cell culture which also rely on oxygen consumption? 

Author Response

Many thanks to Reviewer 1 for the effort made in reviewing my manuscript. With the comments and remarks made, my manuscript will be significantly improved!
Many thanks to Reviewer 1 for the effort made in reviewing my manuscript. With the comments and remarks made, my manuscript will be significantly improved!

All responses to Comments and Remarks of Reviewer 1 are shown in the attach file.

With Respect,
Mitko Petrov
09.08.2021
Sofia, Bulgaria

Reviewer 2 Report

The paper represents a new application of the InterCriteria (decision) Analysis (ICA). The description of the ICA is given correctly and the discussion of the results is exact.

Author Response

Response to Reviewer 2:

Many thanks to the Reviewer for his efforts in reviewing my manuscript. The only note on the manuscript is "Moderate English changes required".

Editing English is the last step in editing the entire manuscript. Before that, I need to respond to the recommendations and comments of other reviewers, which are not few.

With respect:

Mitko Petrov

Reviewer 3 Report

The author presents Modelling and using of Inter-Criteria Decision Analysis for Selecting Growth Rate Models for Batch Cultivation of yeast Kluyveromyces marxianus var. lactis MC 5. This study is interesting. However, as I observed that there are still some corrections as follows:

  1. The introduction is not very useful, therefore, the introduction should be extended very carefully, so, the introduction section should be rewritten again, the introduction should highlight the novelty and motivation of study, not only put some literature without any useful explanation, in fact, the introduction should be clearly stated targets first. Then answer several questions: Why is the topic important (or why do you study on it)? What has been studied? What are your contributions? Why is to propose this particular method?
  2. The literature review (LR) is not presented in a good structure. LR must be moved a new section as Section 2. At the end of LR you should come out with a paragraph to conclude your discussion, in this paragraph you can highlight the novelty of your study also, it means, what the LR done and what you want do. The literature review must highlight the novelty and contribution of the study, but these sections which authors provided only are related works and not literature review, authors must carefully revise these sections.
  3. I advise authors to present the current literature and their contribution to the literature with a summary table.
  4. Some related papers should be mentioned in the manuscript such as: (i) A novel intuitionistic fuzzy MCDM-based CODAS approach for locating an authorized dismantling center: a case study of Istanbul. Waste Management & Research, 38(6), 660-672. (ii) Type-2 neutrosophic number based multi-attributive border approximation area comparison (MABAC) approach for offshore wind farm site selection in USA. Engineering Applications of Artificial Intelligence, 103, 104311.
  5. Research method and other sections:
  • The research approach of this study is not clear.
  • The research strategy of the study is not provided. What kind of strategy did you use for this study? Did you perform any interviews?
  • The data and the procedure of the data collection are not clear.
  1. The flowchart of proposed method must be added in the manuscript.
  2. The authors need to discuss about the limitations of the proposed method as well as limitations, what are your recommendations for future works, how the proposed method solved the the real world problem.

Author Response

Many thanks to Reviewer 3 for the effort made in reviewing my manuscript. With the comments and remarks made, my manuscript will be significantly improved!

With Respect

Mitko Petrov

Reviewer 4 Report

The authors presented the results of the assessment of cell growth models using a multi-criteria decision-making approach - Inter-Criteria Decision Making (ICDA). The authors chose eight main and three additional criteria for evaluation.

The results contain scientific novelty and are of interest to readers.

Several suggestions for improving the manuscript.
1. I suggest justifying the choice of a multi-criteria method for assessing these particular models. It is unclear why the authors chose ICDA. Of course, there is no need to review all the MCDM relates to this study. A brief justification for the choice of ICDA will suffice.

2. The ICDA description is accompanied by links to basic theoretical publications on this method [22 - 25]. However, there are more recent studies by the authors of this method, which describe the ICDA methodology used in the presented manuscript. I suggest expanding the references with, for example, DOI: 10.1007/978-3-319-11313-5_10,  DOI: 10.5220/0005888302840291, DOI: 10.1007/978-3-319-11313-5_9.

3. The list of selected criteria requires a brief justification. The authors refer to the study [21] when choosing the composition of the criteria. However, the cited study also lacks such a rationale.

4. The introduction of additional criteria C9-C11 is not obvious. I suggest describing these criteria, as well as comment in more detail in the manuscript on the need for their introduction into the multi-criteria model.

Author Response

Many thanks to Reviewer 4 for the effort made in reviewing
my manuscript. With the comments and remarks made, my manuscript will be significantly improved!

Comments and Remarks on Reviewer 4:

  1. I suggest justifying the choice of a multi-criteria method for assessing these particular models. It is unclear why the authors chose ICDA. Of course, there is no need to review all the MCDM relates to this study. A brief justification for the choice of ICDA will suffice.
  2. The ICDA description is accompanied by links to basic theoretical publications on this method [22-25]. However, there are more recent studies by the authors of this method, which describe the ICDA methodology used in the presented manuscript. I suggest expanding the references with, for example, DOI: 10.1007/978-3-319-11313-5_10, DOI: 10.5220/0005888302840291, DOI: 10.1007/978-3-319-11313-5_9.

Answer: Thank you very much for the recommendations and comments made by the Reviewer. In the edited version of the manuscript, in Introduction, I briefly mentioned some of the most commonly used methods for MCDM.

The main theoretical publications [22-25] on which the ICDA method is built have been moved to Introduction after the first mention of ICDA. That seems more logical to me. Also, after the theoretical part, a brief justification is made as to why this particular method was used for MCDM.

The suggested publications of the colleagues are cited in the text of the manuscript in Background of ICDA method and are included in the literature.

Thus, I consider that I have responded to the first two comments and Remarks of the Reviewer.

  1. The list of selected criteria requires a brief justification. The authors refer to the study [21] when choosing the composition of the criteria. However, the cited study also lacks such a rationale.

Answer: The criteria for evaluating the parameters in the models were used to validate and verify the adequacy of the models. Statistical criteria - a relative error that shows the error between the experimental and simulated results and the multiple correlation coefficient, which is one of the most commonly used to assess the adequacy of regression models.

Two criteria were used to test the zero hypothesis: statistics l is the criteria for testing the hypothesis of zero mean deviation of the model and experimental data. This criterion has been used in Giridhar and Srivastava [1]. This publication cites Bard, who recommends that this criterion be used to assess the reliability of biotechnological process models. In addition to this criterion for testing the null hypothesis, the standard Fisher coefficient is included.

In the cited literature [21] only the dependences (formulas) for calculating the accepted criteria for model validation are given.

Of course, to test the zero hypothesis, other such criteria may be included. For example: Studen's criteria, Pearson's criteria, etc.

  1. The introduction of additional criteria C9-C11 is not obvious. I suggest describing these criteria, as well as comment in more detail in the manuscript on the need for their introduction into the multi-criteria model.

Answer: The introduction of additional criteria is necessary because in models (15)-(20) we have not two but three kinetic variables – biomass concentration (X), lactose concentration (S), and oxygen concentration (C). A detailed explanation is already included in the manuscript.

I hope that I have responded comprehensively to the recommendations and remarks made by the Reviewer.

With Respect,

Mitko Petrov

09.08.2021

Sofia, Bulgaria

Round 2

Reviewer 1 Report

Accepted in current form

Reviewer 3 Report

All issues have been successfully addressed by authors. So it can be accepted from my side.